# Greatwall-Endosulfine: A Molecular Switch that Regulates PP2A/B55 Protein Phosphatase Activity in Dividing and Quiescent Cells

**DOI:** 10.3390/ijms20246228

**Published:** 2019-12-10

**Authors:** Natalia García-Blanco, Alicia Vázquez-Bolado, Sergio Moreno

**Affiliations:** Instituto de Biología Funcional y Genómica, CSIC, University of Salamanca, 37007 Salamanca, Spain; nataliagb93@usal.es (N.G.-B.); uxiavb@usal.es (A.V.-B.)

**Keywords:** cell cycle, cell growth, CDK, Cyclin, PP2A/B55, Greatwall, ENSA, mitosis, TOR

## Abstract

During the cell cycle, hundreds of proteins become phosphorylated and dephosphorylated, indicating that protein kinases and protein phosphatases play a central role in its regulation. It has been widely recognized that oscillation in cyclin-dependent kinase (CDK) activity promotes DNA replication, during S-phase, and chromosome segregation, during mitosis. Each CDK substrate phosphorylation status is defined by the balance between CDKs and CDK-counteracting phosphatases. In fission yeast and animal cells, PP2A/B55 is the main protein phosphatase that counteracts CDK activity. PP2A/B55 plays a key role in mitotic entry and mitotic exit, and it is regulated by the Greatwall-Endosulfine (ENSA) molecular switch that inactivates PP2A/B55 at the onset of mitosis, allowing maximal CDK activity at metaphase. The Greatwall-ENSA-PP2A/B55 pathway is highly conserved from yeast to animal cells. In yeasts, Greatwall is negatively regulated by nutrients through TORC1 and S6 kinase, and couples cell growth, regulated by TORC1, to cell cycle progression, driven by CDK activity. In animal cells, Greatwall is phosphorylated and activated by Cdk1 at G2/M, generating a bistable molecular switch that results in full activation of Cdk1/CyclinB. Here we review the current knowledge of the Greatwall-ENSA-PP2A/B55 pathway and discuss its role in cell cycle progression and as an integrator of nutritional cues.

## 1. Introduction

Cell cycle progression is driven by the periodic activation and inactivation of Cdk/cyclin complexes. Oscillations in Cdk/cyclin activity depend on the phosphorylation status of the CDK and on cyclin levels [1]. According to the quantitative model of the cell cycle [2], low CDK/cyclin activity in G1 is crucial for the assembly of DNA pre-replication complexes, whereas intermediate and high levels of Cdk/cyclin activity are required to trigger DNA replication and mitosis, respectively [3]. However, this classical Cdk/cyclin-centered model has to take into consideration that protein phosphatases also play an important role in cell cycle regulation [4,5,6,7,8]. Net phosphorylation of Cdk/cyclin substrates depends on the balance of activities between the Cdk/cyclin and its counteracting phosphatases [8]. Specifically, in fission yeast and animal cells, the activity of the protein phosphatase 2A subcomplex PP2A/B55 plays a crucial function in cell cycle control, since it is the main antagonist of Cdk/cyclin phosphorylation [9]. Interestingly, recent findings have shown that substrate-specific Cdk/cyclin activity is also important for cell cycle progression [10]. The Cdk/cyclin substrates contributes by defining the order in which they are phosphorylated, since good Cdk/cyclin substrates tend to be phosphorylated early in the cell cycle, with lower Cdk/cyclin activity, whereas poor Cdk/cyclin substrates require a higher level of Cdk/cyclin activity and are phosphorylated later in the cell cycle [10]. In particular, the nature of the phosphorylated residue and the surrounding amino acid sequence seem to affect the phosphorylation timing of the Cdk/cyclin substrates. Consequently, CDKs prefer serine, while the PP2A/B55 phosphatase prefers threonine residues. As a result, serines are phosphorylated early and threonines are phosphorylated late in the cell cycle [11,12].

Reversible protein phosphorylation is a crucial regulatory mechanism in cell biology [13]. PP2A protein phosphatases are metalloenzymes that catalyze dephosphorylation in a single step using a metal-activated water molecule or hydroxide ion [14], and is a major serine/threonine phosphatase involved in many essential cellular functions [15,16,17]. The PP2A core enzyme consists of a catalytic (C) subunit and a scaffold (A) subunit that associate with a variable regulatory subunit (B) to form the PP2A holoenzyme [18,19]. Interestingly, numerous recent studies have highlighted the role of PP2A in cell cycle control [5,8]. In particular, mitotic entry is the result of the balance between Cdk1/CyclinB kinase activity and the PP2A phosphatase activity in a complex with the regulatory subunit B55 [20,21]. In G2, the PP2A/B55 protein phosphatase complex dephosphorylates Cdk1/CyclinB substrates, opposing Cdk1/CyclinB activity and delaying entry into mitosis until Cdk1/CyclinB activity levels increase above a certain threshold [9]. The Greatwall-Endosulfine module behaves as a molecular switch that inactivates PP2A/B55 at the onset of mitosis [20,21]. Greatwall (Gwl—also known as Mastl in mammals, Rim15 in budding yeast, and Ppk18 and Cek1 in fission yeast) is a member of the AGC family of protein kinases [22,23] that triggers the phosphorylation of Endosulfine, two small proteins in animal cells (ENSA and ARPP-19) and in budding yeast (Igo1 and Igo2) and a single protein in fission yeast (Igo1), which when phosphorylated by Greatwall become potent and specific inhibitors of PP2A/B55 [24,25]. Greatwall was first described in *Drosophila*, where its absence produces defects in chromosome condensation and in mitotic and meiotic progression leading to sterility [26,27]. Further studies showed that the Greatwall-Endosulfine module is conserved in *Xenopus*, where it is required for entry into mitosis [28]; in mammals, where Mastl inactivation causes mitotic and cytokinesis defects [29,30]. Moreover, work in yeast has revealed that the pathway is also conserved in these organisms. In both budding and fission yeasts, nutrients regulate the Greatwall-Endosulfine module. In *Saccharomyces cerevisiae*, TORC1 and protein kinase A (PKA) activities inhibit the Greatwall-Endosulfine switch, which is required for meiotic gene expression and survival in the G0 stationary phase [31,32,33,34]. By contrast, in fission yeast, the Greatwall-Endosulfine switch is negatively regulated by TORC1 [35], and it regulates cell size at division, entry into quiescence [36], and the sexual differentiation response [37,38].

In this review, we will revisit the current knowledge on the role of the PP2A/B55 complex, the main phosphatase that reverse Cdk/cyclin phosphorylations in cell cycle regulation. Particularly, we will discuss the regulation of PP2A/B55 by the highly conserved Greatwall-Endosulfine switch, focusing on the mechanism of action and major physiological implications.

## 2. Regulation of the G2/M Transition by the Greatwall-ENSA-PP2A/B55 Pathway

Cdk/cyclin activity peaks in mitosis, drops during the metaphase to anaphase transition, is low in G1, increases in late G1 to promote entry into S-phase, and continues to rise to very high levels at the end of G2 to induce mitotic onset [2] (Figure 1, blue line). Entry into mitosis is triggered by phosphorylation of multiple Cdk1/CyclinB substrates [39,40]. In early G2, Wee1 phosphorylation of Cdk1 on the tyrosine 15 (Cdk1-Y15) residue keeps Cdk1/CyclinB inactive [41]. During G2, Cdc25 phosphatase levels increase [42,43], counteracting the Cdk1-Y15 phosphorylation by Wee1. At the G2/M transition, the Wee1-Cdc25 balance shifts towards Cdk1-Y15 dephosphorylation by Cdc25 due to a Cdk1 auto-activation loop, where Cdk1/CyclinB phosphorylates both Cdc25 and Wee1, resulting in Cdc25 activation and Wee1 inhibition. As a consequence, Cdk1/CyclinB is fully active, and cells enter mitosis. Nevertheless, the G2/M transition is also controlled by the PP2A/B55 phosphatase complex, the main antagonist of Cdk1/CyclinB activity [9]. In interphase, PP2A/B55 also regulates the Cdk1-Y15 phosphorylation state by promoting Cdc25 dephosphorylation and inactivation [44], as well as Wee1 dephosphorylation and activation [45], ensuring low Cdk1/CyclinB activity levels in G2, and therefore preventing entry into mitosis [9]. Consequently, at the end of G2, PP2A/B55 must be inhibited in order to promote mitotic entry and reach maximal phosphorylation of Cdk1/CyclinB substrates at M-phase. The Greatwall-Endosulfine pathway is responsible for the inactivation of PP2A/B55 at the onset of mitosis. Greatwall kinase phosphorylates Endosulfine proteins, ARPP-19 and ENSA, at a serine residue located at the conserved “FDSpGDY” motif, triggering their binding to PP2A/B55 and the inhibition of the phosphatase complex [24,25]. Interestingly, although ARPP-19 and ENSA are targets of PP2A/B55, their dephosphorylation rate is orders of magnitude lower than Cdk1-phosphorylated substrates. Hence, when ENSA and ARPP-19 are phosphorylated, they show high affinity for the PP2A/B55 complex and compete with Cdk1/CyclinB phosphorylated substrates for the active site of PP2A/B55, preventing their dephosphorylation by “unfair competition” [46].

Different studies have shown that defects in the Greatwall-Endosulfine switch cause cell cycle phenotypes. Accordingly, Greatwall depletion in *Xenopus* egg extracts promotes mitotic exit by inducing dephosphorylation of mitotic substrates, even in the presence of high levels of Cdk1/CyclinB activity [47]. Furthermore, experiments in *Drosophila* have shown that mutations in the *Gwl* gene present a delay in nuclear envelope breakdown (NEB) and anaphase progression, as well as defects in chromosome condensation [26,27]. Work in mammals has also revealed that Mastl-depleted cells display G2 phase delay, slow chromosome condensation, chromosome mis-alignment and mis-segregation, and cytokinesis defects [29,30]. Interestingly, these defects were rescued by chemical inhibition or depletion of PP2A/B55 [29,30,48]. In *Xenopus* and mammalian cells, depletion of ARPP-19 or ENSA also produces defects in mitotic progression [24,25]. Moreover, the ENSA orthologue in *Drosophila*, Endos, is required for proper spindle assembly and oocyte maturation [49,50,51,52]. Endos-deficient cells arrest in prometaphase I and display little phosphorylation of mitotic substrates, consistent with PP2A/B55 being active upon entry into M-phase [50]. In mammalian cells, this pathway is not only involved in G2/M transition and in mitosis because ENSA depletion elongates the duration of S-phase due to lower levels of Treslin, which results in a reduced number of replication forks. This effect goes via PP2A/B55 because treatment with a PP2A/B55 inhibitor, okadaic acid, rescues the effects of ENSA-depletion on Treslin levels [53].

In summary, all these studies confirm that the Greatwall-Endosulfine-PP2A/B55 signaling cascade plays a fundamental role in cell division by regulating the phosphorylation state of Cdk1/CyclinB substrates.

### 2.1. The Greatwall-Endosulfine Switch Inactivates PP2A/B55 at Mitotic Entry

PP2A/B55 protein phosphatase activity fluctuates during the cell cycle with the opposite phase to Cdk1/CyclinB activity, being high in interphase and low in mitosis [6,9,25] (Figure 1, orange line). PP2A/B55 activity oscillation is regulated by the Greatwall-Endosulfine molecular switch, as the Greatwall protein kinase activity is also tightly controlled throughout the cell cycle. Greatwall kinase activity is low in interphase, peaks at mitotic onset, and decreases at the metaphase–anaphase transition [28,30,54]. At the G2/M transition, Greatwall is activated by Cdk1/CyclinB-dependent phosphorylation at two threonine residues (T194 and T207 in humans, or T193 and T206 in *Xenopus*) located in its kinase domain. This phosphorylation triggers Greatwall autophosphorylation at its C-terminal domain (S875 in humans, or S883 in *Xenopus*), resulting in full activation of the kinase [22,23] (Figure 2). In *Drosophila*, Cdk1/CyclinB phosphorylation also mediates Greatwall relocalization from the nucleus to the cytosol by its association to exportin-1 (CRM1) [55]. Greatwall is phosphorylated by Polo kinase, which results in binding to 14–3–3Ɛ proteins and cytosolic retention [56]. In mammals, Mastl is also exported from the nucleus in a CRM1-dependent manner before nuclear envelope breakdown (NEB) [48]; in *Xenopus* cells, importin proteins are responsible for Gwl relocalization [57]. Cytosolic Greatwall phosphorylates ARPP-19 and ENSA proteins, triggering their binding to and inhibition of PP2A/B55. Consequently, inhibition of Cdk1/CyclinB complex by Wee1 phosphorylation at Y15 is relieved, mitotic substrates are phosphorylated, and M-phase is initiated. In *S. cerevisiae*, TORC1 promotes the sequestration of inactive Rim15 by 14–3-3 proteins in the cytoplasm. A drop in TORC1 activity allows the release of Rim15 and its activation, ensuing PP2A/B55 inhibition [58,59]. 

### 2.2. Regulation of the Mitotic Exit by the Greatwall-ENSA-PP2A/B55 Pathway

Mitotic exit requires the reactivation of PP2A/B55 protein phosphatase in order to revert CDK target phosphorylation. At the metaphase/anaphase transition, Greatwall activity is downregulated by dephosphorylation in two successive steps (Figure 3). First, in early anaphase, degradation of CyclinB reduces Cdk1/CyclinB activity levels, which leads to the dephosphorylation of protein phosphatase 1 (PP1) on its Cdk1/CyclinB phosphorylation site at the C-terminus. Dephosphorylation and reactivation of PP1 partially inactivates Greatwall by promoting its dephosphorylation of the S875 residue (in Mastl), which in turn partially releases ENSA and ARPP-19 inhibition of PP2A/B55 [60,61,62]. This is sufficient to promote Greatwall dephosphorylation on T194 and S875 residues by PP2A/B55 phosphatase activity [60,61,62], resulting in full inhibition of Greatwall kinase activity and in its translocation to the nucleus [55]. Recent studies have also described an additional layer of phosphatase regulation of Greatwall-Endosulfine because the RNA polymerase II C-terminal domain phosphatase (Fcp1) dephosphorylates Greatwall, ENSA, and ARPP-19 [63,64]. However, further work will be needed to better comprehend the effects of Fcp1 on the activity of the Greatwall-Endosulfine module.

Collectively, all this data confirm that mitotic entry is governed by two interlinked bistable switches [65,66]. The first bistable switch consists in the inhibition of Wee1 kinase and the activation of Cdc25 phosphatase, which results in dephosphorylation of the Cdk1-Y15 residue, a critical step in the activation of the Cdk1/CyclinB complex. Interestingly, both Wee1 and Cdc25 are Cdk1 substrates, creating a positive feedback loop [67,68,69]. The activation of the Greatwall-Endosulfine module that inactivates PP2A/B55 generates the second bistable switch [66]. The Greatwall kinase is activated by Cdk1/CyclinB-dependent phosphorylation, which is reversed by PP1 and PP2A/B55 phosphatases, showing that both mechanisms are interlinked. Consequently, both systems mutually inhibit each other to ensure that the two switches occupy opposite states (Cdk1 ON, PP2A/B55 OFF and vice versa), providing a robust solution for switch-like mitotic substrate phosphorylation and avoiding futile cycles of target phosphorylation-dephosphorylation [65,66].

## 3. The Greatwall-ENSA-PP2A/B55 Pathway in Yeasts: Nutritional Control of PP2A/B55 Activity

TORC1 monitors the nutritional status of the cell and promotes cell growth. In budding yeast and mammalian cells, cell growth and cell division is coordinated mostly in G1, at START [70,71]. By contrast, in fission yeast, this coordination occurs mainly in G2, at the onset of mitosis [72]. Interestingly, in both budding and fission yeasts, the Greatwall-Endosulfine molecular switch is conserved and is negatively regulated by the TOR pathway. In budding yeast, there is a single Greatwall kinase, encoded by the *RIM15* gene [73,74], whereas Endosulfine is encoded by two genes, *IGO1* and *IGO2* [34]. In fission yeast, there are two Greatwall kinases, encoded by *ppk18* and *cek1*, and a single gene encoding Endosulfine, *igo1* [35]. In nutrient-rich media, Greatwall is phosphorylated and inhibited by the S6 kinase (Sch9 in budding yeast and Sck2 in fission yeast), a conserved downstream target of the TORC1 complex. In nutrient-poor medium, TORC1 and S6 kinase activities drop and the Greatwall kinases Rim15 (in budding yeast) as well as Ppk18 and Cek1 (in fission yeast) become activated, leading to the phosphorylation of Endosulfine and the inhibition of PP2A/B55 activity [35,75].

### 3.1. Regulation of the G1/S Transition 

In budding yeast, the Greatwall-Endosulfine-PP2A/B55 pathway coordinates the nutritional environment to the G1/S transition of the cell cycle by regulating the stability of Sic1, a G1-specific CDK inhibitor, and the expression of G1 cyclins (Figure 4). Sic1 is regulated by phosphorylation at multiple sites. Phosphorylation of residues near the *n*-terminal, carried out by Cdk/cyclins, targets Sic1 for degradation, whereas phosphorylation of Thr173 by the stress-activated MAP kinase Mpk1 stabilizes the protein [76]. Thr173 phosphorylation is reversed by PP2A/CDC55 (PP2A/B55) phosphatase. It has been described that Rim15 activity is not only TORC1-dependent but also cell-cycle-regulated. During G1, there is a peak of Rim15 activity that then drops and remains constant for the rest of the cell cycle [77]. This high activity of Rim15 in G1 phosphorylates Igo1, which inhibits PP2A/CDC55, preventing the removal of Sic1-Thr173 phosphorylation [76]. This phosphorylation also acts as a docking site for Cks1 complexes, Clb5/Cdk1/Cks1. Binding of these complexes prevents the phosphorylation of Sic1 at the *n*-terminus, protecting it from degradation and, at the same time, turning Sic1 into an inhibitor of Clb5/Cdk1/Cks1 [78]. In late G1, an increase in Cdk1/Cln activity results in the initial phosphorylation of Sic1 at Thr5 and Thr33 [79], which primes Sic1 for subsequent phosphorylation at nearby residues. These new phosphorylated residues allow the binding of Sic1 to Cdc4 and the ubiquitination of the protein by SCF^Cdc4^ [79,80], which targets Sic1 for degradation [78]. The G1/S transition depends on Cdk1 activity. The association of Cln3, an early G1 cyclin, with Cdk1 promotes the expression of late G1 cyclins: Cln1 and Cln2. Transcription of CLN1 and CLN2 is regulated by the SCB-binding factor (SBF; E2F in mammalian cells), which is inhibited by Whi5 (Rb in mammalian cells). Interestingly, Whi5 activity is cell-cycle-regulated. In early G1, when Cdk1/Cln3 activity is low, upregulation of Rim15-Igo1,2 prevents the dephosphorylation of Whi5 by PP2A/CDC55, leading to the accumulation of inactive Whi5. In late G1, expression of the G1 cyclins by SBF leads to an increase in Cdk1 activity that further phosphorylates and inactivates Whi5, making START irreversible [77]. 

### 3.2. Regulation of the G2/M Transition

In fission yeast, PP2A/B55 activity limits the rate of mitosis entry. As in animal cells, mitotic onset in fission yeast is the result of the subtle balance between Cdk1/CyclinB protein kinase and PP2A/B55 protein phosphatase activities [20,21]. In *S. pombe*, TORC1 modulates entry into mitosis and cell size at division by inhibiting the Greatwall-Endosulfine switch [35,81,82]. In nitrogen-rich medium, TORC1 is fully active, promoting the inhibition of Greatwall through the activation of the Sck2 S6 kinase (Figure 5). This inhibition of Greatwall keeps PP2A/B55 active, which counteracts Cdk1/CyclinB activity. As a consequence, mitotic entry is delayed, and cells divide with a large size. On the contrary, in nitrogen-poor medium, TORC1 activity drops leading to Greatwall activation, Endosulfine phosphorylation and PP2A/B55 inhibition. Reduced PP2A/B55 activity enables cells to enter mitosis with low levels of Cdk1/CyclinB activity, and thus, cells divide with a smaller size. These results indicate that the Greatwall-Endosulfine pathway integrates nutritional signals and coordinates cell growth (TORC1) with the cell cycle machinery by regulating the activity of PP2A/B55. They also explain why fission yeast cells growing in nitrogen-rich media divide with a larger size compared with fission yeast cells growing in nitrogen-poor media. These changes in cell size also have an impact on cell cycle distribution because cells growing in nitrogen-poor media are advanced into mitosis; as a consequence, they have a short G2 and have to extend the G1 phase of the following cell cycle in order to reach a minimal cell size required to enter S-phase.

### 3.3. PP2A/B55 Connects TORC1 and TORC2 and Provides a Switch from Cell Proliferation to Cell Differentiation

In *S. pombe*, TORC1 and TORC2 complex play opposite roles, TORC1, which is activated by nitrogen, promotes cell growth, and antagonizes cell differentiation, a process that requires TORC2 activity. Inactivation of TORC1 triggers the mating response [83,84], while deletion of TORC2 or its target Gad8 (the fission yeast orthologue of mammalian Akt) causes sterility [85]. Recently, it has been shown that the transition from cell growth to cell differentiation in response to nitrogen starvation, is regulated by PP2A/B55 phosphatase activity [38] through the dephosphorylation of Gad8 at Ser546 [37,82,86]. As indicated above, in low nitrogen, TORC1-Sck2 inactivation leads to the activation of the Greatwall-Endosulfine switch and therefore the inhibition of PP2A/B55, resulting in the accumulation of active Ser546 phosphorylated Gad8 and induction of the differentiation response. Accordingly, in nitrogen-depleted medium cells deleted for *ppk18* and *cek1,* or *igo1*, display a reduction in the frequency of mating and sporulation [35], while deletion of *pab1*, encoding the PP2A B55 subunit, is hyperfertile due to enhanced Gad8 activity [37]. The crosstalk between TORC1 and TORC2 complexes through Greatwall-Endosulfine-PP2A/B55 pathway and Gad8 phosphorylation might explain their opposite roles in the regulation of sexual differentiation response in fission yeast. In nitrogen-rich medium, TORC1 represses the expression of genes required for mating, meiosis, and sporulation, whereas TORC2 promotes the mating response.

In budding yeast, TORC1 and PKA downregulate the differentiation response, which also requires the activation of the Greatwall-Endosulfine-PP2A/B55 pathway, since cells lacking RIM15 or IGO1/2 (Greatwall and Endosulfine orthologues, respectively) show defects in premeiotic S-phase and gametogenesis [87]. Therefore, in both fission and budding yeasts, Greatwall and Endosulfine play important roles in sexual differentiation by downregulating PP2A/B55 activity.

### 3.4. The Greatwall-Endosulfine-PP2A/B55 Pathway Is Required for G1 Arrest and Quiescence under Nitrogen Starvation

When *S. pombe* cells are deprived of nitrogen, they undergo two rounds of cell division, which results in a reduction in cell size, and eventually, G1 arrest [88,89]. If they meet a partner of the opposite mating type or can switch the mating type (homothallic *h^90^*), they initiate the sexual differentiation response, which is followed by meiosis and sporulation; otherwise, if they are heterothallic, they enter a differentiated G0-like state, called quiescence [90]. The establishment of this dormant state takes about 24 hours and is accompanied by a loss of cell polarity, a reduction of cell size, and a flattening of chromatin [90]. During quiescence, cells remain metabolically active and remain viable for months. Nonetheless, upon nitrogen replenishment, cells are able to reenter the mitotic cycle and proliferate again [90,91].

Recently, it has been shown that fission yeast cells lacking Greatwall (*ppk18∆ cek1∆*) or Endosulfine (*igo1∆*) are unable to arrest in G1 under nitrogen starvation, showing only a slight size reduction [35,36]. In fission yeast, two genes, *ppa1* and *ppa2*, encode the catalytic subunit of the PP2A phosphatase, and one gene, *pab1*, the B55 regulatory subunit [92,93]. Consistent with a negative role of PP2A/B55 in the nitrogen starvation response, cells lacking *ppa1* or *ppa2*, or those with reduced levels of *pab1*, can arrest more readily in G1 [35,36]. Furthermore, lowering the expression of *pab1* or deleting the *ppa2* gene rescue the G1 arrest defect of cells lacking *igo1* [36]. All these data suggests that the fission yeast Greatwall-Endosulfine module promotes G1 arrest by inhibiting PP2A/B55 activity upon nitrogen starvation. Similar experiments have also highlighted the importance of the Greatwall-Endosulfine-P22A/B55 pathway for the proper establishment of the G0 phase, as abolishment of the Greatwall-dependent phosphorylation of Igo1, or *igo1* deletion, shows defects in entry into quiescence [36,94]. Moreover, these phenotypes are partially rescued by deletion of the *pab1* gene. The importance of the pathway for meiosis and quiescence is not exclusive to fission yeast. In budding yeast, PP2A/B55 is required for cell survival during quiescence and meiosis [31,32,33,34]. It seems that the function of the PP2A/B55 phosphatase is restricted to early stages of meiosis, because Igo1-S64 phosphorylation increases at the beginning of meiosis and then disappears [31,87].

Interestingly, it has been known for a long time that dietary restriction and downregulation of TORC1 activity prolongs lifespan in diverse organisms, including yeast, flies, worms, fish, rodents, and monkeys [95,96]. Deletion of S6 kinase orthologue (Sch9 in budding yeast and Sck2 in fission yeast), which negatively regulates Greatwall in yeast, also extends the chronological lifespan (CLS) [97,98]. In budding yeast, the Rim15-Igo1/2 pathway is also required for G0 entry, survival in the stationary phase, and the extension of CLS [34,99]. Rim15 in cooperation with other kinases, such as Yak1 (glucose-sensing pathway) and Mck1 (positive regulator of meiosis and sporulation), mediates the accumulation of storage carbohydrates and limits the level of the ROS of cells entering quiescence induced by glucose starvation [100]. It has recently been described that Rim15 is implicated in the glucose-anabolic pathway. It is required not only for the synthesis of trehalose and glycogen that needs to accumulate during entry into quiescence, but also for the accumulation of the β-glucans cell wall. Recently it was reported that sake strains that show an increase in fermentation are deleted for the *RIM15* gene. The lack of Rim15 causes a defective entry into quiescence, an increase in fermentation rate, and a decrease in the level of β-glucans, trehalose, and glycogen during sake fermentation [101,102]. Interestingly, deletion of the *CDC55* gene suppresses the phenotype of high fermentation in a *RIM15* deletion mutant [103].

## 4. Conclusions

The importance of regulating PP2A/B55 protein phosphatase activity in different cellular processes has come to light in recent years. PP2A/B55 and the proteins regulating its activity, Greatwall-Endosulfine, are highly conserved in eukaryotes: from yeast (Rim15-Igo1/2 in budding yeast, Ppk18/Cek1-Igo1 in fission yeast), *Drosophila*, and *Xenopus* (Greatwall-ENSA) to mammalian cells (Mastl-ENSA/Arpp19). Greatwall activity is positively regulated by CDK and negatively regulated by TORC1, turning this pathway into a molecular switch that connects cell growth with the cell cycle. Under nutritional stress, activation of the Greatwall-Endosulfine switch inhibits PP2A/B55 in order to allow completion of the cell cycle, proper G1 arrest, and entry into quiescence.

For the future, there are pending questions to answer related to the role of Greatwall and Endosulfine in the regulation of PP2A/B55 in dividing and quiescent cells, in particular the role of this pathway in cell survival under nutritional stress conditions. It remains possible that TORC1 may also be regulating the activity of Greatwall-Endosulfine in multicellular organisms, where this pathway may be coupling cell growth and cell cycle progression.

## Figures and Tables

**Figure 1 ijms-20-06228-f001:**
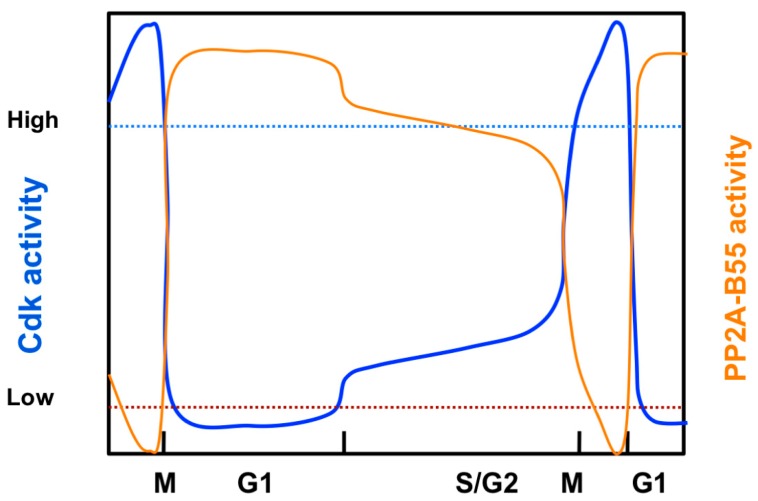
Cdk/cyclin and PP2A/B55 activities fluctuate during the cell cycle with opposite phases. Cdk/cyclin activity (blue line) is low in G1, increases to intermediate levels in S-phase, and peaks during metaphase. On the contrary, PP2A/B55 activity (orange line) increases in anaphase and remains high in G1; it decreases to intermediate levels in S-phase and drops to very low levels at the G2/M transition. Cdk/cyclin activity, through the Greatwall-Endosulfine pathway, downregulates PP2A/B55 activity. This accentuates the switch-like behavior of the system: when the kinase is active, the opposing phosphatase is inactive and vice versa, avoiding futile cycles of phosphorylation–dephosphorylation.

**Figure 2 ijms-20-06228-f002:**
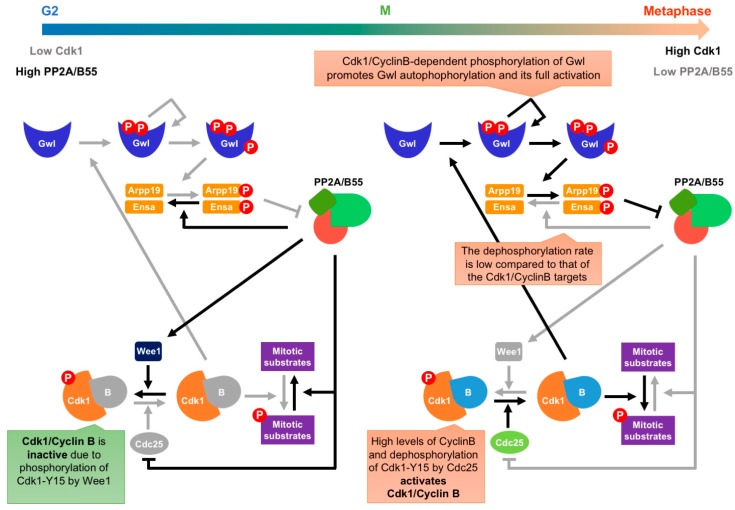
Regulation of the G2/M transition by the Greatwall-Ensa-PP2A/B55 pathway. In early G2, Cdk1/CyclinB activity is low due to low levels of CyclinB and inhibitory Cdk1-Y15 phosphorylation by Wee1. On the contrary, PP2A/B55 activity is high because the Greatwall-Endosulfine switch is off and unable to inhibit PP2A/B55. This phosphatase regulates Cdk1-Y15 phosphorylation state by inhibiting Cdc25 and activating Wee1. PP2A/B55 also antagonizes Cdk1/CyclinB activity by dephosphorylating mitotic substrates. As cells progress through G2, CyclinB and Cdc25 protein levels increase and Cdk1/CyclinB activity rises and activates the Greatwall-Endosulfine switch, resulting in inhibition of PP2A/B55 activity. Low activity of PP2A/B55 releases the inhibition of the Cdk1/CyclinB.

**Figure 3 ijms-20-06228-f003:**
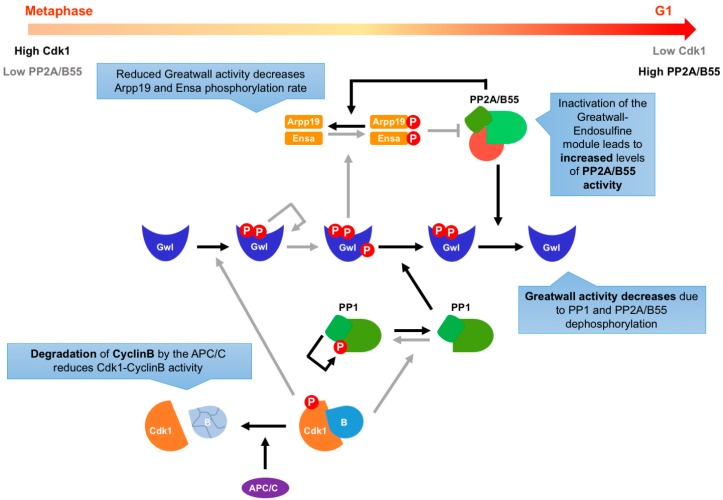
Regulation of mitotic exit by the Greatwall-Ensa-PP2A/B55 pathway. To successfully exit mitosis, the high Cdk1/CyclinB activity has to be downregulated. In early anaphase, CyclinB degradation reduces the activity levels of the Cdk1/CyclinB complex, which allows the dephosphorylation of protein phosphatase 1 (PP1). PP1 is then able to partially dephosphorylate Greatwall. This dephosphorylation promotes the activation of PP2A/B55, which dephosphorylates other residues of Greatwall, inhibiting its activity. When the Greatwall-Endosulfine switch is OFF, PP2A/B55 activity increases and reverses the phosphorylation of multiple CDK substrates during mitotic exit.

**Figure 4 ijms-20-06228-f004:**
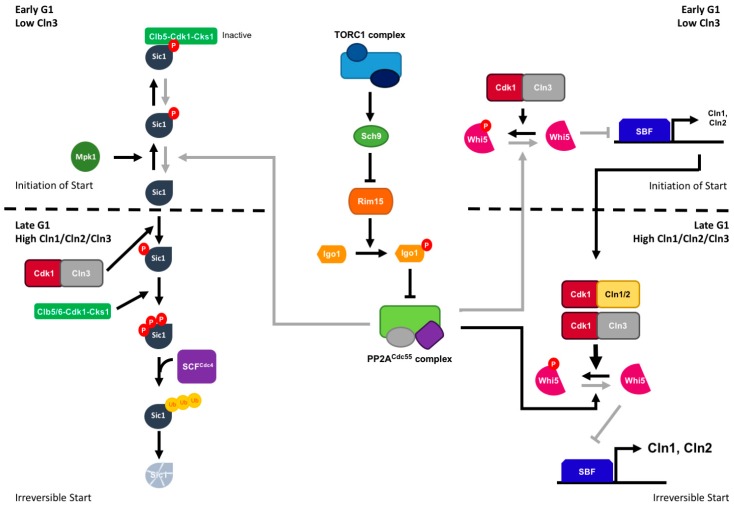
Regulation of the G1/S transition by the Greatwall-Endosulfine-PP2A/Cdc55 pathway in budding yeast. In early G1, Rim15 activity peaks resulting in inhibition of the PP2A/Cdc55 phosphatase. As a result, Sic1 is phosphorylated at Thr173 by Mpk1. This phosphorylation stabilizes the Sic1 protein, which interacts with Clb5/CKS/Cks1 and inactivates the Cdk1/Clb5. This drop in phosphatase activity also helps maintaining Whi5 in a phosphorylated and inactive state, which allows for transcription of Cln1 and Cln2 cyclins. As cells progress through G1, the activity of Rim15 decreases, and the PP2A/B55 phosphatase activity rises, causing the dephosphorylation of Sic1. Dephosphorylated Sic1 can then be phosphorylated first by Cln-CDK and then by Clb5,6/CDK/Cks1. CDK-dependent phosphorylation promotes the ubiquitylation and degradation of Sic1. The increase in Cln2 and Cln3 synthesis results in further Whi5 phosphorylation and inactivation by the CDK-Cyclin complexes, leading to the activation of the SBF transcription factor and START becomes irreversible.

**Figure 5 ijms-20-06228-f005:**
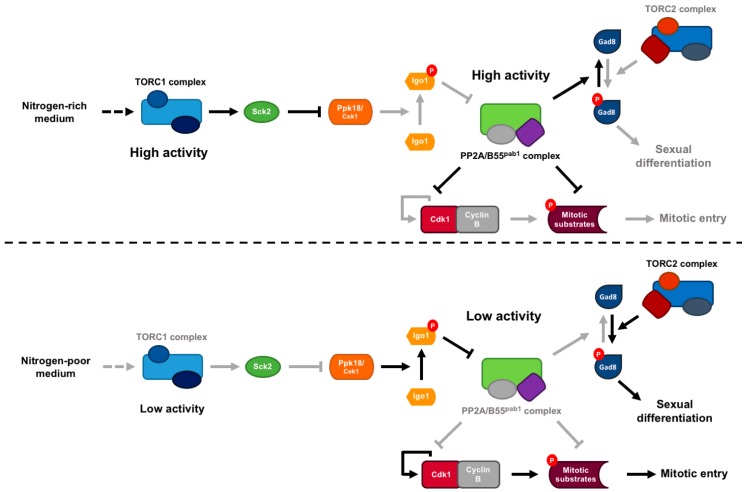
Nutritional regulation of the cell size and cell differentiation by the TORC1-Greatwall-Ensa-PP2A/B55 pathway in fission yeast. In nitrogen-rich medium, the TORC1 complex is highly active. In this condition, the Ppk18/Cek1-Igo1 switch is inhibited by the fission yeast S6 kinase homologue Sck2. Therefore, the activity of the PP2A/B55 complex is high and counteracts the phosphorylation of the mitotic substrates carried out by the Cdk1/CyclinB, so mitosis is delayed, and cells divide with a large size. High PP2A/B55 phosphatase also inhibits the phosphorylation of Gad8 and the sexual differentiation response. When cells are shifted to a nitrogen-poor medium, TORC1 complex activity drops, and the Ppk18/Cek1-Igo1 module is activated and inhibits PP2A/B55. Low activity of PP2A/B55 allows entry into mitosis with reduced levels of Cdk1-Cyclin B activity, and with a smaller cell size. The phosphatase also no longer inhibits Gad8 phosphorylation; as a result, the sexual differentiation response is induced.

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
