# Peer review of "Greatwall-Endosulfine: A Molecular Switch that Regulates PP2A/B55 Protein Phosphatase Activity in Dividing and Quiescent Cells"

_ijms, 2019, doi:10.3390/ijms20246228_

Round 1

Reviewer 1 Report

In this review article, Sergio Moreno and his colleagues describe the regulation of the Greatwall-Endosulfine-PP2A:B55 pathway based on the most recent information. The review covers an enormous amount of information and it is well-written although a little bit wordy because of repetitive statements.

I only have minor suggestions:

Abstract (line 22-23): The sentence is too complicated: ‘In animal cells, Greatwall is 21 phosphorylated and activated by Cdk1 at G2/M generating a bistable molecular switch that activates 22 Cdk1/CyclinB at metaphase.’ and Cdk1 is activated earlier than metaphase.

Line 80: Cdk activity drops during meta- and anaphase.

Line 115: Gwl mutation cause NEB ??? A loss of function mutation should delay NEB.

Line 228-229: Whi5 phosphorylation marks Start rather than S-phase initiation. S-phase starts when Sic1 is degraded and Clb5:Cdk1 is activated.

Line251: better: ‘……inhibition of Gwl maintains PP2A/B55 activity.’ rather than activation.

Author Response

I only have minor suggestions:

Abstract (line 22-23): The sentence is too complicated: ‘In animal cells, Greatwall is 21 phosphorylated and activated by Cdk1 at G2/M generating a bistable molecular switch that activates 22 Cdk1/CyclinB at metaphase.’ and Cdk1 is activated earlier than metaphase.

Lines 22-23: We have changed this sentence to: “In animal cells, Greatwall is phosphorylated and activated by Cdk1 at G2/M, generating a bistable molecular switch that results in full activation of Cdk1/CyclinB”.

Line 80: Cdk activity drops during meta- and anaphase.

Line 83: We have corrected this sentence to: “Cdk/cyclin activity peaks in metaphase, drops during the metaphase to anaphase transition, is low in G1,…”

Line 115: Gwl mutation cause NEB ??? A loss of function mutation should delay NEB.

This reviewer is correct! Gwl mutants show delayed NEB. We have corrected this sentence (please see lines 121-122).

Line 228-229: Whi5 phosphorylation marks Start rather than S-phase initiation. S-phase starts when Sic1 is degraded and Clb5:Cdk1 is activated.

We have also corrected this (please see lines 246-253).

Line 251: better: ‘……inhibition of Gwl maintains PP2A/B55 activity.’ rather than activation.

Line 272: This sentence has been corrected.

Reviewer 2 Report

Very interesting manuscript on regulation of PP2A/B55 protein phosphatase activity. Very innovative topic considering Greatwall-Endosulfine pathways. Manuscript is well written and I did not found any errors. I would recommend to accept the manuscript.

Author Response

Thank you very much for your very nice comment!

Reviewer 3 Report

This is an interesting review of the roles and regulation of the Greatwall-Endosulfine-PP2A/B55 pathway in cell division and proliferation control. The emphasis on the fission and budding yeasts is well deserved, as the literature too often gives little space to work in these important models. In addition, the findings regarding the Gwl-PP2A pathway can be difficult to reconcile between animals and yeasts. The literature can become confusing, and for this reason, this review is very useful. Here are some suggestions for improvements before publication.

Line 66: It would be good to cite Archambault et al, 2007, PLoS Gen, for defects in meiosis in Drosophila gwl mutant.

Line 68: At least one reference should be cited for work in Xenopus, ideally Yu et al, 2006, Mol Cell.

Line 101: The other essential element of the unfair competition model is the extremely high affinity of pENSA for PP2A/B55.

Figure 1: Where do these curves come from? PP2A/B55 has been shown to reactivate only minutes after Cdk1 is inactivated. This is because PP2A/B55 takes time to free itself from the inhibition by pENSA/pARPP19. This is important to order the events of mitotic exit. See Cundell et al, 2013, Mol Cell. It would be careful to draw a much simpler model, or to leave it out completely.

For the reason just explained above, I also disagree with this sentence on lines 133-134: “PP2A/B55 protein phosphatase activity fluctuates during the cell cycle, being high in interphase and low in mitosis [25], exactly with the opposite phase to Cdk1/CyclinB activity (Figure 1).”

Lines 119-120: “…indicating that they were caused by improper dephosphorylation of Cdk1/CyclinB substrates…”. The results do not indicate that the important substrates are of Cdk1/Cyclin B. PP2A-B55 can dephosphorylate substrates of other kinases.

Lines 122-123: ENSA is called Endos in Drosophila and it is also required for correct mitosis (Rangone et al, 2011, PLoS Gen).

Line 143: “Once in the cytoplasm, Greatwall is phosphorylated by Polo kinase…” It has not been shown that this occurs in the cytoplasm. It could also occur in the nucleus as Polo enters the nucleus at the G2/M transition and Gwl shuttles between nucleus and cytoplasm.

Figure 2: I think the arrow from Cdk1/B to Gwl should be from the active, unphosphorylated Cdk1/B. By the way, figures are truncated on the right.

Figure 3: It is inaccurate to state that “Reduced Greatwall activity decreases Arpp19 and Ensa binding to and inhibition of PP2A/B55”. According to the prevailing model, all that reducing Gwl activity does is it reduces the phosphorylation rate of Arpp19 and Ensa. This combined with the continued dephosphorylation of Arpp19 and Ensa by PP2A/B55 results in a shift of the pools towards unphosphorylated forms, and PP2A/B55 then becomes increasingly available to dephosphorylate its other substrates.

Figure 3 makes us wonder what dephosphorylates PP1.

Section 2.2: It could be added that PP2A/B55 dephosphorylates Gwl to promote its return to the nucleus (Wang et al, 2016, Cell Cycle). It would also be good to provide a brief overview of the known effector proteins dephosphorylated by PP2A/B55 in the late events of mitotic exit, including in central spindle function (ex: PRC1) and nuclear envelope reformation (ex: BAF).

It might be interesting to include a short section to discuss how functions of the Gwl-ENSA-PP2A/B55 module may have evolved to have different importance in difference systems. Meanwhile, an apparently unique or different function in one system can also contribute in a more minor way in other systems. One example is how the Rim15-Igo1/2-PP2A-Cdc55 contributes to promote mitotic entry in budding yeast. This could be briefly reviewed.

Author Response

Thank you very much for the positive review. We have introduced the suggested references and modified the manuscript according to the reviewers comments.

Line 66: It would be good to cite Archambault et al, 2007, PLoS Gen, for defects in meiosis in Drosophila gwl mutant.

Line 68: We have cited this reference (27).

Line 68: At least one reference should be cited for work in Xenopus, ideally Yu et al, 2006, Mol Cell.

Line 70: We have introduced this reference (28).

Line 101: The other essential element of the unfair competition model is the extremely high affinity of pENSA for PP2A/B55.

Thanks for the comment, we have introduced this on lines 106-107.

Figure 1: Where do these curves come from? PP2A/B55 has been shown to reactivate only minutes after Cdk1 is inactivated. This is because PP2A/B55 takes time to free itself from the inhibition by pENSA/pARPP19. This is important to order the events of mitotic exit. See Cundell et al, 2013, Mol Cell. It would be careful to draw a much simpler model, or to leave it out completely.

The curves come from data published in Coudreuse and Nurse 2010. Nature 468: 1074-1079 for the CDK activity and in Mochida et al 2010. Science 330: 1670-1673 (Fig. 3C) and Mochida and Hunt 2012. EMBO Rep. 13:197-2013 (Fig. 2A) for the PP2A/B55 activity. Based on this and on data from our group published in Chica et al. 2016. Curr. Biol. 26: 319-330, Prof. Bela Novak generated a mathematical model (see Figures S5B and S5D in Chica et al. 2016), which is in agreement with the idea that the activities of Cdk1 and PP2A/B55 oscillate with opposite phases during the cell cycle.

For the reason just explained above, I also disagree with this sentence on lines 133-134: “PP2A/B55 protein phosphatase activity fluctuates during the cell cycle, being high in interphase and low in mitosis [25], exactly with the opposite phase to Cdk1/CyclinB activity (Figure 1).”

In line 146-147, we have removed the word “exactly” and changed the sentence to: “PP2A/B55 protein phosphatase activity fluctuates during the cell cycle with the opposite phase to Cdk1/CyclinB activity, being high in interphase and low in mitosis [6,9,25](Figure 1)”

Lines 119-120: “…indicating that they were caused by improper dephosphorylation of Cdk1/CyclinB substrates…”. The results do not indicate that the important substrates are of Cdk1/Cyclin B. PP2A-B55 can dephosphorylate substrates of other kinases.

Line 125: We agree with the reviewer and have removed this sentence in the revised version of the manuscript.

Lines 122-123: ENSA is called Endos in Drosophila and it is also required for correct mitosis (Rangone et al, 2011, PLoS Gen).

Thanks for the correction. On lines 127 and 128, we have used Endos instead of ENSA and introduced the suggested reference (52).

Line 143: “Once in the cytoplasm, Greatwall is phosphorylated by Polo kinase…” It has not been shown that this occurs in the cytoplasm. It could also occur in the nucleus as Polo enters the nucleus at the G2/M transition and Gwl shuttles between nucleus and cytoplasm.

Line 156: We have corrected this and have removed “Once in the cytoplasm”

Figure 2: I think the arrow from Cdk1/B to Gwl should be from the active, unphosphorylated Cdk1/B. By the way, figures are truncated on the right.

Figure 2: Thanks. We have corrected the position of the arrow.

The figures look all right in our pdf. We have noticed that during the generation of the pdf from the word file, the figures are adjusted to the width of the figure legend.

Figure 3: It is inaccurate to state that “Reduced Greatwall activity decreases Arpp19 and Ensa binding to and inhibition of PP2A/B55”. According to the prevailing model, all that reducing Gwl activity does is it reduces the phosphorylation rate of Arpp19 and Ensa. This combined with the continued dephosphorylation of Arpp19 and Ensa by PP2A/B55 results in a shift of the pools towards unphosphorylated forms, and PP2A/B55 then becomes increasingly available to dephosphorylate its other substrates.

Thanks. We have corrected this in figure 3 to “Reduced Greatwall activity decreases Arpp19 and Ensa phosphorylation rate”.

Figure 3 makes us wonder what dephosphorylates PP1.

Thanks. We introduced an arrow in figure 3 indicating the PP1 dephosphorylates itself during the metaphase to anaphase transition when Cdk1/cyclinB activity starts to drop.

Section 2.2: It could be added that PP2A/B55 dephosphorylates Gwl to promote its return to the nucleus (Wang et al, 2016, Cell Cycle).

Lines 186-187: Thanks. We have introduced this comment and the reference (55).

It would also be good to provide a brief overview of the known effector proteins dephosphorylated by PP2A/B55 in the late events of mitotic exit, including in central spindle function (ex: PRC1) and nuclear envelope reformation (ex: BAF).

We feel that the review is already too long to introduce the additional information suggested by the reviewer. We prefer to maintain the focus of the review on the connection between Cdk/cyclin activity and PP2A/B55 through Greattwall and Endosulfine.

It might be interesting to include a short section to discuss how functions of the Gwl-ENSA-PP2A/B55 module may have evolved to have different importance in difference systems. Meanwhile, an apparently unique or different function in one system can also contribute in a more minor way in other systems. One example is how the Rim15-Igo1/2-PP2A-Cdc55 contributes to promote mitotic entry in budding yeast. This could be briefly reviewed.

The literature on the role of PP2A-B55 in mitosis in budding yeast is controversial. There is one report indicating that Rim15 and Igo1,2 are required for timely entry into mitosis under temperature stress (Juanes et al.2013. PloS Genetics). However, other reports indicate that PP2A/B55 promotes, rather than prevents, entry into mitosis (Rossio and Yoshida. 2001. J. Cell Biol.), which is the opposite to the role of PP2A/B55 in animal cells and fission yeast. We have avoided to discuss these papers until this confusion is resolved.

Reviewer 4 Report

This manuscript by Garcia-Blanco et al., reviewed up-to-date knowledge of Greatwall-Ensa pathway on cell cycle regulation and cellular response to nutritional changes. The text was clearly written and figures were prepared well. Therefore the manuscript will be of great interest to researchers in this field as well as to general readers of IJMS. 

I have some specific comments/suggestions for improvement as follows:

1. Right sides of Figures 2-5 are not visible (torn?) in the pdf downloaded from the web site. The figure size should be optimized.

2. The authors can refer to Sajiki et al. Sci Adv. 2018 in 3.4. (p.10) where phenotypes of Endosulfine deletion mutant are discussed (p.10, l.308-310).

Author Response

Thanks for your general comment!

I have some specific comments/suggestions for improvement as follows:

1. Right sides of Figures 2-5 are not visible (torn?) in the pdf downloaded from the web site. The figure size should be optimized.

The figures look all right in our pdf. We have noticed that during the generation of the pdf from the word file, the figures were adjusted to the width of the figure legend.

2. The authors can refer to Sajiki et al. Sci Adv. 2018 in 3.4. (p.10) where phenotypes of Endosulfine deletion mutant are discussed (p.10, l.308-310).

We have cited this paper on line 344 (reference 94).